# Some Results on the Symmetric Representation of the Generalized Drazin Inverse in a Banach Algebra

**Yonghui Qin [1], Xiaoji Liu [2,\*] and Julio Benítez [3,\*]** 

[1] College of Mathematics and Computing Science, Guilin University of Electronic Technology, Guilin 541004, China; yonghui1676@163.com

[2] College of Mathematics and Computer Science, Guangxi University for Nationalities, Nanning 530006, China

[3] Department of Applied Mathematics, Universitat Politècnica de València, 46022 Valencia, Spain

\* Correspondence: xiaojiliu72@126.com (X.L.); jbenitez@mat.upv.es (J.B.)

**Abstract:** Based on the conditions $ab^2 = 0$ and $b^\pi(ab) \in \mathscr{A}^d$, we derive that $(ab)^n$, $(ba)^n$, and $ab + ba$ are all generalized Drazin invertible in a Banach algebra $\mathscr{A}$, where $n \in \mathbb{N}$ and $a$ and $b$ are elements of $\mathscr{A}$. By using these results, some results on the symmetry representations for the generalized Drazin inverse of $ab + ba$ are given. We also consider that additive properties for the generalized Drazin inverse of the sum $a + b$.

**Keywords:** generalized Drazin inverse; Banach algebra; representation

## 1. Introduction

Let $\mathscr{A}$ be a complex unital Banach algebra with unit 1. The sets of all invertible elements and quasinilpotent elements of $\mathscr{A}$ are denoted by $\mathscr{A}^{-1}$ and $\mathscr{A}^{\mathrm{qnil}}$, respectively, where $\mathscr{A}^{-1} := \{a \in \mathscr{A} : \exists\, x \in \mathscr{A} : ax = xa = 1\}$ and $\mathscr{A}^{\mathrm{qnil}} := \{a \in \mathscr{A} : \lim_{n \to +\infty} \|a^n\|^{1/n} = 0\}$. Let $a \in \mathscr{A}$ and, if there is a element $b \in \mathscr{A}$ such that

$$bab = b, \qquad ab = ba, \text{ and} \qquad a(1 - ab) \text{ is quasinilpotent,} \tag{1}$$

then $b$ is the *generalized Drazin inverse* of $a$, denoted by $a^d$, and it is unique. The set of generalized Drazin-invertible elements is denoted by $\mathscr{A}^d = \{a \in \mathscr{A} : \exists\, a^d\}$. In particular, if $a(1 - ab) = 0$ (or $aba = a$), then $b$ is called the *group inverse* of $a$. Note that $aa^d$ is an idempotent element and let $a^\pi = 1 - aa^d$. It was given, in [1] (Lemma 2.4), that $a^d$ exists if and only if there is an idempotent $q \in \mathscr{A}$, such that $aq = qa$, $aq$ is quasinilpotent, and $a + q$ is invertible.

The generalized inverse in a matrix or operator theory is very useful in scientific calculation and in various engineering technologies [2–4]. It is well known that the Drazin inverse has been applied in a few fields, such as statistics and probability [5], ordinary differential equations [6], Markov chains [7], operator matrices [8], neural network models [9,10], and the references therein. In [11], a study of the Drazin inverse for bounded linear operators in a Banach space $X$ is given, when 0 is an isolated spectral point of the operator. In [12], some additive results on the Drazin inverse, under the condition $ab = 0$, are obtained. However, as in [12,13], this condition was not enough to derive a formula for the generalized Drazin inverse for $a + b$. In [14], authors investigated how to express the Drazin inverse of sums, differences, and products of two matrices $P$ and $Q$, under the conditions $P^3Q = QP$ and $Q^3P = PQ$. The representations of the Drazin inverse for $(P + Q)$, such that $PQP = 0$ and $PQ^2 = 0$, is

given in [15]. The generalized inverses in $C^*$-algebras has been investigated in [16] and a symmetry of the generalized Drazin inverse in a $C^*$-algebra has been considered in [17].

Some additive properties of the generalized Drazin inverse in a Banach algebra were investigated in [18]. Recently, the expression for the generalized Drazin inverse of the sum $a + b$ on Banach algebra has been studied, such as in the representations of the generalized Drazin inverse for $a + b$ in a Banach algebra, obtained in [19]; some new additive results for the generalized Drazin inverse in a Banach algebra, given in [20]; and additive results on the generalized Drazin inverse of a sum of two elements in a Banach algebra are derived in [21] and the references therein. In this paper, we consider the representations of the generalized Drazin inverse of the sum of two elements in a Banach algebra. By using the assumed conditions $ab^2 = 0$ and $b^\pi(ab) \in \mathscr{A}^{\mathrm{d}}$, it is implied that $(ab)^n$, $(ba)^n$, and $ab + ba \in \mathscr{A}^{\mathrm{d}}$, and a symmetry representation for the generalized Drazin inverse of $ab + ba$ is obtained, where $n \in \mathbb{N}$ and $a, b \in \mathscr{A}^{\mathrm{d}}$. We also consider the additive properties for the generalized Drazin inverse of the sum $a + b$.

In Section 2, some notation is introduced and lemmas are given. In Section 3, a symmetric representation of the generalized Drazin inverse for $ab + ba$ in a Banach algebra is derived. The additive properties of the generalized Drazin inverse of $a + b$ are investigated in Section 4.

## 2. Preliminaries

Let $\mathscr{B}$ be a subalgebra of the unital algebra $\mathscr{A}$. For an element $b \in \mathscr{B}^{-1}$, the inverse of $b$ in $\mathscr{B}$ is denoted by $[b^{-1}]_{\mathscr{B}}$. As in [19], it is given that $\mathscr{B}^{-1} \not\subset \mathscr{A}^{-1}$. Let $\mathscr{P} = \{p_1, p_2, \dots, p_n\}$ be a *total system of idempotents* in $\mathscr{A}$ if $p_i^2 = p_i$, for all $i$, $p_i p_j = 0$ if $i \neq j$, and $p_1 + \cdots + p_n = 1$, as in [22]. If $a \in \mathscr{A}^{\mathrm{d}}$, then

$$a = \begin{bmatrix} a_1 & 0 \\ 0 & a_2 \end{bmatrix}_{\mathscr{P}}, \qquad a^{\mathrm{d}} = \begin{bmatrix} [a_1^{-1}]_{p\mathscr{A}p} & 0 \\ 0 & 0 \end{bmatrix}_{\mathscr{P}}, \qquad a^\pi = \begin{bmatrix} 0 & 0 \\ 0 & 1-p \end{bmatrix}_{\mathscr{P}}, \tag{2}$$

where $p = aa^{\mathrm{d}}$, $\mathscr{P} = \{p, 1-p\}$, $a_1 \in [p\mathscr{A}p]^{-1}$, and $a_2 \in [(1-p)\mathscr{A}(1-p)]^{\mathrm{qnil}}$. If $a$ has the representation given as in (2), then $a^{\mathrm{d}} = [a_1^{-1}]_{p\mathscr{A}p} = a_1^{\mathrm{d}}$.

The following lemmas are required in what follows.

**Lemma 1** ([19])**.** *Let $\mathscr{P} = \{p, 1-p\}$ be a total system of idempotents in $\mathscr{A}$, and let $a, b \in \mathscr{A}$ have the following representation*

$$a = \begin{bmatrix} x & 0 \\ z & y \end{bmatrix}_{\mathscr{P}}, \qquad b = \begin{bmatrix} x & t \\ 0 & y \end{bmatrix}_{\mathscr{P}}.$$

*Then there exist $(z_n)_{n=0}^\infty \subset (1-p)\mathscr{A}p$ and $(t_n)_{n=0}^\infty \subset p\mathscr{A}(1-p)$, such that*

$$a^n = \begin{bmatrix} x^n & 0 \\ z_n & y^n \end{bmatrix}_{\mathscr{P}}, \qquad b^n = \begin{bmatrix} x^n & t_n \\ 0 & y^n \end{bmatrix}_{\mathscr{P}}, \qquad \forall\, n \in \mathbb{N}.$$

**Lemma 2** ([22])**.** *Let $a, b \in \mathscr{A}$ be generalized Drazin invertible and $ab = 0$. Then, $a + b$ is generalized Drazin invertible and*

$$(a+b)^{\mathrm{d}} = b^\pi \sum_{n=0}^\infty b^n (a^{\mathrm{d}})^{n+1} + \sum_{n=0}^\infty (b^{\mathrm{d}})^{n+1} a^n a^\pi.$$

**Lemma 3** ([22])**.** *Let $x, y \in \mathscr{A}$, $p$ be an idempotent of $\mathscr{A}$, and let $x$ and $y$ have the representation*

$$x = \begin{bmatrix} a & 0 \\ c & b \end{bmatrix}_{\{p, 1-p\}}, \qquad y = \begin{bmatrix} b & c \\ 0 & a \end{bmatrix}_{\{1-p, p\}}. \tag{3}$$

(i) *If $a \in [p\mathscr{A}p]^{\mathsf{d}}$ and $b \in [(1-p)\mathscr{A}(1-p)]^{\mathsf{d}}$, then $x, y \in \mathscr{A}^{\mathsf{d}}$ and*

$$x^{\mathsf{d}} = \begin{bmatrix} a^{\mathsf{d}} & 0 \\ u & b^{\mathsf{d}} \end{bmatrix}_{\{p,1-p\}}, \qquad y^{\mathsf{d}} = \begin{bmatrix} b^{\mathsf{d}} & u \\ 0 & a^{\mathsf{d}} \end{bmatrix}_{\{1-p,p\}}, \tag{4}$$

*where*

$$u = \sum_{n=0}^{\infty} (b^{\mathsf{d}})^{n+2} c a^n a^{\pi} + \sum_{n=0}^{\infty} b^{\pi} b^n c (a^{\mathsf{d}})^{n+2} - b^{\mathsf{d}} c a^{\mathsf{d}}. \tag{5}$$

(ii) *If $x \in \mathscr{A}^{\mathsf{d}}$ and $a \in [p\mathscr{A}p]^{\mathsf{d}}$, then $b \in [(1-p)\mathscr{A}(1-p)]^{\mathsf{d}}$, and $x^{\mathsf{d}}$ and $y^{\mathsf{d}}$ are given by (4) and (5).*

**Lemma 4** ([11]). *Let $a \in \mathscr{A}^{\mathsf{d}}$. Then $[(a)^n]^{\mathsf{d}} = [a^{\mathsf{d}}]^n$, for all $n = 1, 2, \cdots$.*

**Lemma 5** ([11]). *If $a, b \in \mathscr{A}^{\mathsf{d}}$ and $ab = ba = 0$. Then, $(a+b)^{\mathsf{d}}$ also exists and $(a+b)^{\mathsf{d}} = a^{\mathsf{d}} + b^{\mathsf{d}}$.*

**Lemma 6** ([23]). *Let $a, b \in \mathscr{A}^{\mathsf{d}}$. Then $(ab)^{n+1}$ is generalized Drazin invertible, for some $n \in \mathbb{N}$, if and only if $ab$ is generalized Drazin invertible.*

**Lemma 7** ([23]). *Let $a, b \in \mathscr{A}^{\mathsf{d}}$ and $(ab)^{n+1}$ be generalized Drazin invertible for some $n \in \mathbb{N}$. Then, $(ba)^n$ is generalized Drazin invertible and $[(ba)^n]^{\mathsf{d}} = b[(ab)^{n+1}]^{\mathsf{d}}a$.*

## 3. The Symmetric Representation for the Generalized Drazin Inverse of $ab + ba$

Let $a, b \in \mathscr{A}^{\mathsf{d}}$. A symmetric expression of $(ab + ba)^{\mathsf{d}}$ is given, by using $ab$, $ba$, $(ab)^{\mathsf{d}}$, and $(ba)^{\mathsf{d}}$, with the following assumed conditions

$$ab^2 = 0, \quad b^{\pi}(ab) \in \mathscr{A}^{\mathsf{d}}. \tag{6}$$

**Theorem 1.** *Let $a, b \in \mathscr{A}^{\mathsf{d}}$ satisfy (6). Then, $(ab)^n, (ba)^n, ab + ba \in \mathscr{A}^{\mathsf{d}}$ ($n = 1, 2, \cdots$), and a representation of $(ab + ba)^{\mathsf{d}}$ is given as*

$$(ab + ba)^{\mathsf{d}} = (ba)^{\pi} \sum_{n=1}^{\infty} (ba)^{n-1} [(ab)^n]^{\mathsf{d}} + \sum_{n=1}^{\infty} [(ba)^n]^{\mathsf{d}} (ab)^{n-1} (ab)^{\pi}. \tag{7}$$

**Proof.** Let $b = \begin{bmatrix} b_1 & 0 \\ 0 & b_2 \end{bmatrix}_{\mathscr{P}}$, where $\mathscr{P} = \{bb^{\mathsf{d}}, b^{\pi}\}$, $b_1$ is invertible in the subalgebra $bb^{\mathsf{d}}\mathscr{A}bb^{\mathsf{d}}$, and

$b_2$ is quasinilpotent. Let us write $a = \begin{bmatrix} a_{11} & a_{12} \\ a_{21} & a_{22} \end{bmatrix}_{\mathscr{P}}$. From $ab^2 = 0$, we have

$$a_{11} = 0, \qquad a_{21} = 0, \qquad a_{12}b_2^2 = 0, \qquad \text{and} \qquad a_{22}b_2^2 = 0. \tag{8}$$

Thus, we have $ab = \begin{bmatrix} 0 & a_{12}b_2 \\ 0 & a_{22}b_2 \end{bmatrix}$. By Lemma 3, we obtain that $ab \in \mathscr{A}^{\mathsf{d}}$ if and only if $a_{22}b_2$ is generalized Drazin invertible. Thus, $(b^{\pi}ab)^{\mathsf{d}}$ exists. By using Cline's formula, it proves that $(ab)^{\mathsf{d}}$ also is. Therefore, we obtain $(ab)^n, (ba)^n \in \mathscr{A}^{\mathsf{d}}$ by using Lemma 6 and 7. Since $ab^2 = 0$, by Lemma 2 we can prove that $ab + ba$ is generalized Drazin invertible and that (7) holds. If $n = 1$, then $(ab + ba)^{\mathsf{d}} = (ba)^{\pi}(ab)^{\mathsf{d}} + (ba)^{\mathsf{d}}(ab)^{\pi}$. By using mathematical induction, we derive that the representation can be given, as in (7). □

**Remark 1.** *Note that the expression given in Theorem 1 is symmetric.*

**Theorem 2.** *Let* $a, b \in \mathscr{A}^d$ *satisfy* (6) *and* $a^2 = 0$. *Then* $ab + ba \in \mathscr{A}^d$ *and* $[(ab + ba)^d]^n = [(ab)^d]^n + [(ba)^d]^n$, *for all* $n = 1, 2, \cdots$.

**Proof.** Let $a, b$ be written as in the proof of Theorem 1, and, by $ab^2 = 0$, we derive $ab = \begin{bmatrix} 0 & a_{12}b_2 \\ 0 & a_{22}b_2 \end{bmatrix}$

and $ab, ba, (ab)^n, (ba)^n \in \mathscr{A}^d$. Since $ab^2 = 0$ and $a^2 = 0$, we have

$$(ab)^n(ba)^n = (ba)^n(ab)^n = 0, \quad (ab + ba)^n = (ab)^n + (ba)^n, \tag{9}$$

for all $n = 1, 2, \cdots$. By Lemma 4, Lemma 5, and the first equality of (9), we derive

$$[(ab + ba)^d]^n = [(ab + ba)^n]^d = [(ab)^n + (ba)^n]^d = [(ab)^n]^d + [(ba)^n]^d = [(ab)^d]^n + [(ba)^d]^n.$$

□

At the end of Section 3, let $\mathscr{A}$ be a $C^*$-algebra, as in [17]. Then, a simple application of the generalized Drazin inverse in a $C^*$-algebra can be given, as follows.

**Theorem 3.** *Let* $a, b \in \mathscr{A}$ *be group invertible. If* (6) *is satisfied, then* $(ab + ba)^\dagger$ *exists.*

**Proof.** By using Theorem 1, we derive that $ab + ba$ is group invertible. As pointed out in [16], $ab + ba$ is generalized invertible. Thus, $(ab + ba)^\dagger$ exists. □

**Theorem 4.** *Let* $a, b \in \mathscr{A}^d$. *If* (6) *is satisfied, then* $(ab + ba)^d$ *is self-adjoint in a* $C^*$-*algebra.*

**Proof.** Note that $ab + ba$ is self-adjoint in a $C^*$-algebra. By Theorem 1 and using [17] (Theorem 3.2), we obtain that $(ab + ba)^d$ is self-adjoint in a $C^*$-algebra. □

## 4. The Representation for the Generalized Drazin Inverse of $a + b$

In this section, we consider some results on the expression of $(a + b)^d$, by using $a$, $b$, $a^d$, and $b^d$, where $a, b \in \mathscr{A}^d$.

**Lemma 8.** *Let* $a, b \in \mathscr{A}^d$ *satisfy* $ab^2 = 0$. *Then,* $(a + b)^d$ *exists if and only if* $b^\pi(a + b) \in \mathscr{A}^d$.

**Proof.** Similarly, we rewrite $a, b$ as in the proof of Theorem 1. Since $ab^2 = 0$, we derive

$$a + b = \begin{bmatrix} b_1 & a_{12} \\ 0 & b_2 + a_{22} \end{bmatrix}_{\mathscr{P}}. \tag{10}$$

By Lemma 3, note that $(a + b)^d$ exists if and only if $(a_{22} + b_2)^d$ exists; that is, $(a + b)^d$ exists if and only if $b^\pi(a + b)$ is generalized Drazin invertible. □

**Theorem 5.** *Let* $a, b \in \mathscr{A}^d$ *satisfy the conditions of Theorem 2. Then*

$$(a + b)^d = \sum_{n=0}^{\infty} (b^d)^{2n+1} \left[ b^d(ab)^\pi(ab)^n a + (ab)^\pi(ab)^n \right]$$

$$- \sum_{n=0}^{\infty} b^\pi b^{2n} \left\{ [(ab)^d]^{n+1} a + b[(ab)^d]^{n+1} \right\}.$$

**Proof.** By Lemma 8, it also leads to (10). By Lemma 3, we can prove that $(a + b)^d$ exists if and only if $(a_{22} + b_2)^d$ exists; that is, $(a + b)^d$ exists if and only if $b^\pi(a + b)$ is generalized Drazin invertible.

If $b^\pi ab \in \mathscr{A}^{\mathrm{d}}$, then $(a_{22}b_2)^{\mathrm{d}}$ exists. By Cline's formula, we have that $(b_2a_{22})^{\mathrm{d}}$ exists. As in the proof of Theorem 1, by Lemma 6 and 7, we also obtain that $(ab)^n, (ba)^n \in \mathscr{A}^{\mathrm{d}}$, for all $n = 1, 2, \cdots$.

By $a^2 = 0$, we get

$$a_{12}a_{22} = 0 \qquad \text{and} \qquad a_{22}^2 = 0. \tag{11}$$

By (8) and (11), we have $(b_2a_{22})(a_{22}b_2) = 0, \ (a_{22}b_2)(b_2a_{22}) = 0$. Using Lemma 5, and by Cline's formula, we derive

$$(a_{22}b_2 + b_2a_{22})^{\mathrm{d}} = (a_{22}b_2)^{\mathrm{d}} + (b_2a_{22})^{\mathrm{d}}. \tag{12}$$

By induction, let $[(a_{22}b_2)^{\mathrm{d}} + (b_2a_{22})^{\mathrm{d}}]^n = [(a_{22}b_2)^{\mathrm{d}}]^n + [(b_2a_{22})^{\mathrm{d}}]^n$ for all $n \geq 1$. Therefore, we can prove that

$$[(a_{22}b_2 + b_2a_{22})^{\mathrm{d}}][(a_{22}b_2)^{\mathrm{d}} + (b_2a_{22})^{\mathrm{d}}]^n = [(a_{22}b_2)^{\mathrm{d}}]^{n+1} + [(b_2a_{22})^{\mathrm{d}}]^{n+1}.$$

Since $(a_{22}b_2 + b_2a_{22})b_2^2 = 0$ and $b_2$ are quasinilpotent, by Lemma 5 and (12), we obtain

$$\begin{aligned}
[(a_{22} + b_2)^2]^{\mathrm{d}} &= (a_{22}b_2 + b_2a_{22} + b_2^2)^{\mathrm{d}} \\
&= \sum_{n=0}^{\infty} b_2^{2n}[(a_{22}b_2 + b_2a_{22})^{\mathrm{d}}]^{n+1} \\
&= \sum_{n=0}^{\infty} b_2^{2n} \left[ (a_{22}b_2)^{\mathrm{d}} + (b_2a_{22})^{\mathrm{d}} \right]^{n+1}.
\end{aligned} \tag{13}$$

Then, $b^\pi(a + b) \in \mathscr{A}^{\mathrm{d}}$ implies that $(a_{22} + b_2)^{\mathrm{d}}$ exists and $(a_{22} + b_2)^{\mathrm{d}} = [(a_{22} + b_2)^2]^{\mathrm{d}}(a_{22} + b_2)$. Finally, by (13), and $(b_2a_{22})^{\mathrm{d}} = b_2 \left[ (a_{22}b_2)^{\mathrm{d}} \right]^2 a_{22}$, we obtain

$$\begin{aligned}
(a_{22} + b_2)^{\mathrm{d}} &= \left[ (a_{22} + b_2)^{\mathrm{d}} \right]^2 (a_{22} + b_2) \\
&= \sum_{n=0}^{\infty} b_2^{2n} \left\{ \left[ (a_{22}b_2)^{\mathrm{d}} \right]^{n+1} + \left( b_2 \left[ (a_{22}b_2)^{\mathrm{d}} \right]^2 a_{22} \right)^{n+1} \right\} (a_{22} + b_2) \\
&= \sum_{n=0}^{\infty} b_2^{2n} \left[ (a_{22}b_2)^{\mathrm{d}} \right]^{n+1} a_{22} + \sum_{n=0}^{\infty} b_2^{2n} \left[ b_2 \left( (a_{22}b_2)^{\mathrm{d}} \right)^2 a_{22} \right]^{n+1} b_2 \\
&= \sum_{n=0}^{\infty} b_2^{2n} \left\{ \left[ (a_{22}b_2)^{\mathrm{d}} \right]^{n+1} a_{22} + b_2 \left[ (a_{22}b_2)^{\mathrm{d}} \right]^{n+1} \right\}
\end{aligned} \tag{14}$$

and

$$(a_{22} + b_2)^\pi = (a_{22}b_2)^\pi - \sum_{n=0}^{\infty} b_2^{2n+1} \left\{ \left[ (a_{22}b_2)^{\mathrm{d}} \right]^{n+1} a_{22} + b_2 \left[ (a_{22}b_2)^{\mathrm{d}} \right]^{n+1} \right\}. \tag{15}$$

By Lemma 3, we get that $a + b \in \mathscr{A}^{\mathrm{d}}$ and

$$(a + b)^{\mathrm{d}} = \begin{bmatrix} b_1^{-1} & u \\ 0 & (a_{22} + b_2)^{\mathrm{d}} \end{bmatrix}_{\mathscr{P}}, \tag{16}$$

and

$$u = \sum_{n=0}^{\infty} (b_1^{-1})^{n+2} a_{12}(b_2 + a_{22})^n (a_{22} + b_2)^\pi - (a_{22} + b_2)^{\mathrm{d}} a_{12} b_1^{-1}. \tag{17}$$

Evidently, we have $\left[b_1^{-1}\right]_{\mathscr{P}} = b^{\mathsf{d}}$ and

$$
b^{\mathsf{d}} ba = \left[ \begin{array}{cc} b_1^{-1} b_1 & 0 \\ 0 & 0 \end{array} \right]_{\mathscr{P}} \left[ \begin{array}{cc} 0 & a_{12} \\ 0 & a_{22} \end{array} \right]_{\mathscr{P}} = \left[ \begin{array}{cc} 0 & a_{12} \\ 0 & 0 \end{array} \right]_{\mathscr{P}} = a_{12}.
$$

One easily has (by induction and by using (8) and (11)) that, if $n \geq 1$, then

$$
a_{12}(a_{22} + b_2)^n = \begin{cases} a_{12}(b_2 a_{22})^{n/2} & \text{if } n \text{ is even,} \\ a_{12}(b_2 a_{22})^{(n-1)/2} b_2 & \text{if } n \text{ is odd.} \end{cases} \tag{18}
$$

By Lemma 1, we obtain that, for any $n \geq 1$,

$$
b^{\pi}(ba)^n = \left[ \begin{array}{cc} 0 & 0 \\ 0 & b^{\pi} \end{array} \right]_{\mathscr{P}} \left[ \begin{array}{cc} 0 & x_n \\ 0 & (b_2 a_{22})^n \end{array} \right]_{\mathscr{P}} = \left[ \begin{array}{cc} 0 & 0 \\ 0 & (b_2 a_{22})^n \end{array} \right]_{\mathscr{P}} = (b_2 a_{22})^n,
$$

where $(x_n)_{n=0}^{\infty}$ is a sequence in $\mathscr{A}$. Furthermore, one has $b_2 = b^{\pi} b = b b^{\pi}$ and $ab^{\pi} = a(1 - bb^{\mathsf{d}}) = a(1 - b^2(b^{\mathsf{d}})^2) = a$. Hence, if $n \geq 1$ is even, then

$$
a_{12}(a_{22} + b_2)^n = a_{12}(b_2 a_{22})^{n/2} = b^{\mathsf{d}} b a b^{\pi}(ba)^{n/2} = b^{\mathsf{d}} ba(ba)^{n/2} = b^{\mathsf{d}}(ba)^{(n+2)/2},
$$

and if $n \geq 1$ is odd, then

$$
a_{12}(a_{22} + b_2)^n = a_{12}(b_2 a_{22})^{(n-1)/2} b_2 = b^{\mathsf{d}} b a b^{\pi}(ba)^{(n-1)/2} b^{\pi} b = b^{\mathsf{d}}(ba)^{(n+1)/2} b.
$$

From (15), we have

$$
\begin{aligned}
a_{12}(a_{22} + b_2)^{\pi} &= a_{12}(1 - b_2(a_{22} b_2)^{\mathsf{d}} a_{22}), \\
a_{22}(a_{22} + b_2)^{\pi} &= (a_{22} b_2)^{\pi} a_{22}, \\
a_{12} b_2(a_{22} + b_2)^{\pi} &= a_{12} b_2 (a_{22} b_2)^{\pi}, \\
a_{22} b_2(a_{22} + b_2)^{\pi} &= a_{22} b_2 (a_{22} b_2)^{\pi}.
\end{aligned}
$$

Thus, by using the obvious equality $(ba)^k b = b(ab)^k$, and by (14)–(16) and (18), we have

$$
\begin{aligned}
(a + b)^{\mathsf{d}} &= b_1^{\mathsf{d}} + u = [b_1]_{\mathscr{P}}^{-1} + \sum_{n=0}^{\infty} \left( \left[b_1^{-1}\right]_{\mathscr{P}} \right)^{n+2} a_{12}(b_2 + a_{22})^n (a_{22} + b_2)^{\pi} \\
&\quad - (a_{22} + b_2)^{\mathsf{d}} a_{12} b_1^{-1} + (a_{22} + b_2)^{\mathsf{d}} \\
&= \sum_{n=0}^{\infty} (b^{\mathsf{d}})^{2n+2} b^{\pi}(ab)^n a + \sum_{n=0}^{\infty} (b^{\mathsf{d}})^{2n+1} b^{\pi}(ab)^n \\
&\quad - \sum_{n=0}^{\infty} b^{\pi} b^{2n} \left\{ [(ab)^{\mathsf{d}}]^{n+1} a + b[(ab)^{\mathsf{d}}]^{n+1} \right\} \\
\\
&= \sum_{n=0}^{\infty} (b^{\mathsf{d}})^{2n+1} \left[ b^{\mathsf{d}}(ab)^{\pi}(ab)^n a + (ab)^{\pi}(ab)^n \right] \\
&\quad - \sum_{n=0}^{\infty} b^{\pi} b^{2n} \left\{ [(ab)^{\mathsf{d}}]^{n+1} a + b[(ab)^{\mathsf{d}}]^{n+1} \right\}.
\end{aligned}
$$

The proof is completed. $\square$

**Theorem 6.** *Let* $a, b \in \mathscr{A}^{\mathsf{d}}$ *satisfy* (6) *and* $b^{\pi} a^2 = 0$. *Then,*

$$(a + b)^{\mathsf{d}} = b^{\mathsf{d}} + u + v,$$

*where*

$$v = -\left\{ b^{\mathsf{d}} a (ba)^{\mathsf{d}} + \sum_{n=0}^{\infty} b^{\mathsf{d}} b^{2n+1} \left[ ((ab)^{\mathsf{d}})^{n+1} + ((ba)^{\mathsf{d}})^{n+1} \right] \right\},$$

$$u = \sum_{n=0}^{\infty} (b^{\mathsf{d}})^{n+2} a (a + b)^n + \sum_{n=0}^{\infty} (1 - b^{\pi}) b^n a v^{n+2} - b^{\mathsf{d}} a v.$$

**Proof.** Let $p = bb^{\mathsf{d}}$ and $\mathscr{P} = \{p, 1 - p\}$. Let $a$ and $b$ have the following representation

$$b = \begin{bmatrix} b_1 & 0 \\ 0 & b_2 \end{bmatrix}_{\mathscr{P}}, \qquad a = \begin{bmatrix} a_3 & a_1 \\ a_4 & a_2 \end{bmatrix}_{\mathscr{P}}, \tag{19}$$

where $b_1$ is invertible in $p\mathscr{A}p$ and $b_2$ is quasinilpotent in $(1 - p)\mathscr{A}(1 - p)$. Let us find the expression of $b^{\pi} a^2$ in the system of idempotents $\mathscr{P}$:

$$b^{\pi} a^2 = \begin{bmatrix} 0 & 0 \\ 0 & 1 - p \end{bmatrix}_{\mathscr{P}} \begin{bmatrix} 0 & a_1 \\ 0 & a_2 \end{bmatrix}_{\mathscr{P}} \begin{bmatrix} 0 & a_1 \\ 0 & a_2 \end{bmatrix}_{\mathscr{P}} = \begin{bmatrix} 0 & 0 \\ 0 & a_2^2 \end{bmatrix}_{\mathscr{P}} = a_2^2.$$

Thus, $a_2^2 = 0$. On the other hand,

$$ab^2 = \begin{bmatrix} 0 & a_1 \\ 0 & a_2 \end{bmatrix}_{\mathscr{P}} \begin{bmatrix} b_1^2 & 0 \\ 0 & b_2^2 \end{bmatrix}_{\mathscr{P}} = \begin{bmatrix} 0 & a_1 b_2^2 \\ 0 & a_2 b_2^2 \end{bmatrix}_{\mathscr{P}}.$$

Therefore, $a_2 b_2^2 = 0$. By $b^{\pi} ab, b^{\pi} ba \in \mathscr{A}^{\mathsf{d}}$, we obtain $(a_2 b_2), (b_2 a_2) \in \mathscr{A}^{\mathsf{d}}$. We can appeal to Theorem 5, obtaining (recall that $b_2$ is quasinilpotent and $b_2^{\mathsf{d}} = 0$) that

$$(a_2 + b_2)^{\mathsf{d}} = -a_2 (b_2 a_2)^{\mathsf{d}} - \sum_{n=0}^{\infty} b_2^{2n+1} \left[ ((a_2 b_2)^d)^{n+1} + ((b_2 a_2)^d)^{n+1} \right].$$

From Lemma 3 and the representation of $a + b$ in (16), we have

$$
\begin{aligned}
(a + b)^{\mathsf{d}} &= \left[ b_1^{-1} \right]_{\mathscr{P}} + (a_2 + b_2)^{\mathsf{d}} + u \\
&= \left[ b_1^{-1} \right]_{\mathscr{P}} + u - \left\{ a_2 (b_2 a_2)^d + \sum_{n=0}^{\infty} b_2^{2n+1} \left[ ((a_2 b_2)^d)^{n+1} + ((b_2 a_2)^d)^{n+1} \right] \right\},
\end{aligned}
\tag{20}
$$

*where*

$$
\begin{aligned}
u &= \sum_{n=0}^{\infty} \left( \left[ b_1^{-1} \right]_{\mathscr{P}} \right)^{n+2} a_1 (a_2 + b_2)^n (a_2 + b_2)^{\pi} \\
&\quad + \sum_{n=0}^{\infty} b_1^{\pi} b_1^n a_1 ((a_2 + b_2)^{\mathsf{d}})^{n+2} - \left[ b_1^{-1} \right]_{\mathscr{P}} a_1 (a_2 + b_2)^{\mathsf{d}} \\
&= \sum_{n=0}^{\infty} (b_1^{\mathsf{d}})^{n+2} a_1 (a_2 + b_2)^n.
\end{aligned}
$$

Observe that $\left[ b_1^{-1} \right]_{\mathscr{P}} = b^{\mathrm{d}}$, and

$$
\begin{aligned}
(b^{\mathrm{d}})^{n+2} a (a+b)^n &= \left[ \begin{array}{cc} (b_1^{\mathrm{d}})^{n+2} & 0 \\ 0 & 0 \end{array} \right]_{\mathscr{P}} \left[ \begin{array}{cc} 0 & a_1 \\ 0 & a_2 \end{array} \right]_{\mathscr{P}} \left[ \begin{array}{cc} b_1^n & x_n \\ 0 & (a_2+b_2)^n \end{array} \right]_{\mathscr{P}} \\
&= \left[ \begin{array}{cc} 0 & 0 \\ 0 & (b_1^{\mathrm{d}})^{n+2} a_1 (a_2+b_2)^n \end{array} \right]_{\mathscr{P}} = (b_1^{\mathrm{d}})^{n+2} a_1 (a_2+b_2)^n, \\
v = b^{\pi} (a+b)^{\mathrm{d}} &= (a_2+b_2)^{\mathrm{d}} = - \left\{ b^{\mathrm{d}} a (ba)^{\mathrm{d}} + \sum_{n=0}^{\infty} b^{\mathrm{d}} b^{2n+1} \left[ ((ab)^{\mathrm{d}})^{n+1} + ((ba)^{\mathrm{d}})^{n+1} \right] \right\}.
\end{aligned}
$$

Thus, the above expression of $u$ reduces to

$$
u = \sum_{n=0}^{\infty} (b^{\mathrm{d}})^{n+2} a (a+b)^n + \sum_{n=0}^{\infty} (1 - b^{\pi}) b^n a (v)^{n+2} - b^{\mathrm{d}} a v. \tag{21}
$$

Expressions (20) and (21) finish the proof. □

## 5. Conclusions

In this paper, we have proved that the multiplications $(ab)^n$ and $(ba)^n$ of elements $a, b \in \mathscr{A}^{\mathrm{d}}$ in a Banach algebra are both generalized Drazin invertible with the conditions (6). A symmetry representation of the generalized Drazin inverse for $ab + ba$ has been derived. The expression given in Theorem 1 is symmetric, as in Remark 1. In the other words, if the result is applied in the computation of $(ab + ba)^{\mathrm{d}}$, maybe it will improve the corresponding computational effectiveness and reduce its complexity. The additive properties of $(a+b)^{\mathrm{d}}$ have been investigated under the conditions $ab^2 = 0$, $b^{\pi} ab \in \mathscr{A}^{\mathrm{d}}$, and $a^2 = 0$. With similar conditions, but $a^2 = 0$ being replaced by $b^{\pi} a^2 = 0$, we have also given a resulting expression of $(a+b)^{\mathrm{d}}$.

In fact, as pointed out as in [19], it is still an interesting and open problem to express the generalized Drazin inverse of $a + b$ as a function of $a$, $b$, and their respective generalized Drazin inverses. In the future, we plan to consider the representations of the generalized Drazin inverse for $a \pm b$ by using $a$, $b$, and their generalized Drazin inverses, without side conditions.

**Author Contributions:** Funding acquisition, Y.Q. and X.L.; Methodology, X.L.; Supervision, J.B.; Writing-review and editing, Y.Q.

**Funding:** This work was supported by the National Natural Science Foundation of China (grant number: 11361009, 61772006,11561015), the Special Fund for Science and Technological Bases and Talents of Guangxi (grant number: 2016AD05050, 2018AD19051), the Special Fund for Bagui Scholars of Guangxi (grant number: 2016A17), the High level innovation teams and distinguished scholars in Guangxi Universities (grant number: GUIJIAOREN201642HAO), the Natural Science Foundation of Guangxi(grant number: 2017GXNSFBA198053, 2018JJD110003), and the open fund of Guangxi Key laboratory of hybrid computation and IC design analysis (grant number: HCIC201607).

**Conflicts of Interest:** The authors declare that they have no conflict of interest.

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
