# Peer review of "Some Results on the Symmetric Representation of the Generalized Drazin Inverse in a Banach Algebra"

_symmetry, doi:10.3390/sym11010105_

Round 1

Reviewer 1 Report

The Authors consider a GDI in Banach algebra with, as they conclude in the Summary 'some new conditions' which are not verbalised explicitly, or not stressed enough by the Authors. In addition, the soundness of the new conditions is not discussed, also from the viewpoint of potential computational effectiveness and complexity. 

Please consider the reference to ON THE GENERALIZED DRAZIN INVERSE AND GENERALIZED RESOLVENT, by Djordjevic and Stanimirovic and also relate the work to other inverse matrix problems, also with Moore-Penrose inverse, if possible. 

Reviewer 2 Report

0.  The paper is interesting and the results are worth of publication  in this specially sure . 

The authors  should be more specific in the abstract and in the conclusions,  about the results explained in the paper :  it is quite pleasant for the readers to understand,  at least partially,  what main results will be obtained in the paper with some details. 

some statement should be improved from an English language point  of view.

For example, section 3:

a. In this section, we consider some symmetric the (???) expression of (ab + ba)d with a, b A d  and (better to use a comma) by using ab, ba, (ab)d, and (ba)d

b. 

Theorem 3.1. Let a,b Ad such that ab2 = 0. If bπ(ab) is generalized Drazin invertible, then ab, ba, (ab)n, (ba)n A d, and ab + ba is (are) generalized Drazin invertible, which (???) an (???) symmetry representation of (ab + ba)d can be given as 

c. 

Theorem 3.2. Let a, b A d such that ab2 = 0 and a2 = 0. If bπ (ab) is generalized Drazin invertible, then ab + ba is generalized Drazin invertible, which (???) an (???) symmetry representation of (ab + ba)d can be given as [(ab+ba)d]n =[(ab)d]n +[(ba)d]n for all n=1,2,···

d. analogous statements 
